# Two Cohorts, One Network: Consensus Master Regulators Orchestrating Papillary Thyroid Carcinoma

**DOI:** 10.3390/ijms262211231

**Published:** 2025-11-20

**Authors:** Diana Tapia-Carrillo, Octavio Zambada-Moreno, Enrique Hernández-Lemus, Hugo Tovar

**Affiliations:** 1Escuela Nacional Preparatoria Plantel 9 “Pedro de Alba”, Universidad Nacional Autónoma de Mexico, Mexico City 07300, Mexico; diana.tapia@enp.unam.mx; 2Deparment of Genetic Engineering, Center for Research and Advanced Studies (Cinvestav), Irapuato 36821, Mexico; octavio.zambadam@cinvestav.mx; 3Computational Genomics Division, National Institute of Genomic Medicine, Mexico City 14610, Mexico

**Keywords:** papillary thyroid carcinoma, transcriptional master regulators, gene regulatory networks, molecular endocrinology, estrogen signaling, TGF-β signaling, meta-analysis

## Abstract

Papillary thyroid carcinoma (PTC) is the most common endocrine malignancy, yet the transcriptional hierarchies linking endocrine signaling to tumor progression remain poorly defined. Here, we integrated gene-expression profiles from two independent cohorts (TCGA-THCA and GSE33630) to identify consensus transcriptional master regulators (TMRs) driving PTC. After normalization and differential expression analysis, we reconstructed regulon networks with ARACNe-AP, inferred TMR activity using VIPER, and integrated evidence across datasets via Fisher’s meta-analysis. This cross-cohort strategy yielded 50 shared TMRs, predominantly from the Zinc Finger, Forkhead, ETS, and nuclear receptor families. Network topology highlighted *PBX4*, *GATAD2A*, *BHLHE40*, *HEY2*, and *TEAD4* as upstream regulators controlling other TMRs. Functional enrichment revealed activation of NOTCH, MAPK, PI3K, and TGF-β signaling and enrichment of early and late estrogen-response programs, uncovering a noncanonical role of *SMAD9* in TGF-β signaling. Together, these findings delineate the transcriptional and hormonal circuitry underlying thyroid tumorigenesis, providing a regulatory framework for biomarker-driven therapies based on network activity states.

## 1. Introduction

Papillary thyroid carcinoma (PTC) is the most common endocrine malignancy worldwide, representing the majority of thyroid cancer cases and accounting for 1–1.5% of all newly diagnosed cancers each year [1,2]. Its incidence has steadily increased over the past decades in most regions, likely due to improved detection and genuine biological trends, while underdiagnosis in low-resource settings may still obscure the true burden [1]. At the molecular level, PTC develops from deregulated gene expression programs driven by oncogenic lesions (e.g., *BRAF*V600E, RAS) and influenced by endocrine and differentiation signals [3,4,5]. Despite extensive research on canonical signaling pathways such as MAPK and PI3K/AKT [6,7,8], the upstream transcriptional regulators that coordinate these tumor programs across patient groups remain incompletely understood.

A major source of this uncertainty is the limited use of integrative, network-level approaches in thyroid cancer. Most studies focus on identifying differentially expressed genes or individual signaling pathways but do not explore which transcription factors (TFs) sit at the top of the regulatory hierarchy and drive tumor-promoting states. Transcriptional master regulators (TMRs) are TFs that control coherent regulons of target genes, influencing differentiation, hormonal responses, invasiveness, and other malignant traits [9,10,11]; identifying them requires going beyond gene lists and reconstructing gene regulatory networks (GRNs) from expression data [12,13,14]. However, in PTC, there remains no comprehensive framework that (i) compares independent cohorts, (ii) infers regulatory hierarchy, and (iii) links TMRs to endocrine signaling and epithelial–mesenchymal transition (EMT) programs—gaps that impede both mechanistic understanding and therapeutic progress [6,15,16].

To address this gap, we conducted a transcriptomic meta-analysis by integrating two independent PTC cohorts (TCGA-THCA papillary adenocarcinoma, Not Otherwise Specified (NOS), and GSE33630) to identify consensus TMRs that are consistently active across datasets. Instead of focusing on individual genes or isolated pathways, we employed a systems-biology approach based on gene regulatory network reconstruction and protein activity inference to reveal regulatory hierarchies that may promote tumor progression.Primary objective:

To identify transcriptional master regulators that are consistently active in PTC across independent cohorts using a cross-platform, network-based meta-analysis framework.


Secondary objectives:
To analyze the hierarchical organization of these master regulators using network topology;To characterize the biological pathways and tumorigenic processes (e.g., NOTCH, MAPK, PI3K/AKT, TGF-β, EMT, cytoskeletal remodeling, estrogen-response programs) under their regulation; andTo assess their potential importance for endocrine crosstalk and therapeutic targeting in PTC.


By explicitly integrating two datasets and enforcing consensus at the regulator level, this study provides a data-driven regulatory map of PTC. It clarifies which TFs are likely to function as network “drivers” in this tumor type and establishes a basis for subsequent experimental validation and biomarker-guided interventions.

## 2. Results and Discussion

A cross-cohort meta-analysis of TCGA-THCA and GSE33630 identified 50 transcriptional master regulators in papillary thyroid carcinoma. These include members of the Zinc Finger, Forkhead, nuclear receptor, and E26 transformation-specific (ETS) families—groups with established roles in proliferation, migration, metastasis, and thyroid differentiation. Using a quantile-based criterion on the ARACNe + motif network (high out-degree with low in-degree; upper-quartile vs. lower-quartile thresholds), the most consistent top-of-cascade candidates were *PBX4*, *GATAD2A*, *BHLHE40*, *HEY2*, and *TEAD4*. Within this framework, we observed the interactions *BHLHE40* → *RUNX2* and *TEAD4* → *SMAD9*, as well as a reciprocal loop between *FOXQ1* and *GRHL3*, indicating hierarchical modules with potential system-level impact.

### 2.1. Meta-Analysis of Master Regulators

The meta-analysis combining VIPER-based master regulator analysis (MRA) from TCGA-THCA and GSE33630 (papillary thyroid carcinoma, adenocarcinoma, NOS) identified 50 TMRs with false discovery rate (FDR)-adjusted Fisher’s combined *p* < 0.05, supported across both cohorts. As shown in Figure 1, many transcription factors did not reach strong significance; however, this set of 50 TMRs also showed positive mean normalized enrichment score (NES) values (greater than 1), emphasizing their role as key regulators in PTC.

Forkhead (FOX) family TMRs. We identified three TMRs from the Forkhead family. Notably, *FOXE1* is a well-characterized regulator of thyroid development and differentiation; its dysregulation has been linked to thyroid tumorigenesis and metastatic progression through modulation of the epithelial–mesenchymal transition via *ZEB1* [17,18,19]. We also detected *FOXP2*, which has been implicated in tumor establishment, with overexpression associated with increased proliferation, migration, evasion of apoptosis, metastasis, and immune cell infiltration. In our analysis, *FOXQ1* was overexpressed, a pattern reported to correlate with worse prognosis in thyroid cancer patients.

Nuclear receptor TMRs. We identified four transcription factors from the nuclear receptor superfamily—*RARA*, *RARB*, *RXRG*, and *ESRRG* (also known as ERRγ). This finding is clinically significant because retinoids have been shown to induce redifferentiation of thyroid cells and restore radioiodine uptake, although responses vary; in this context, RARβ/RXRγ expression has been linked to increased treatment sensitivity in thyroid cancer cell lines and tumors [20,21,22]. Additionally, ERRγ (*ESRRG*) has emerged as a promising drug target, as ERRγ inverse agonists boost sodium–iodide symporter (NIS) expression and improve radioiodine uptake in models of papillary and anaplastic thyroid carcinoma [23,24,25]. This TMR hub and its regulons provide a mechanistic framework for their role in disease, highlighting potential therapeutic opportunities.

ETS family. TMRs include *ETV1*, *ETV4*, and *ETV5*—members of the ETS family—which are involved in activating the canonical MAPK pathway in PTC [26]. Their presence among the top 50 TMRs aligns with disease biology. Functionally, *ETV4* promotes proliferation and invasion, while *ETV5* mediates BRAF^V600E signaling and drives growth and epithelial–mesenchymal transition, highlighting their roles as pro-oncogenic nodes [4,27,28]. Additionally, 15 other transcription factor families are represented in our analysis, each with fewer TMRs; the mapping between the 50 TMRs and their families is summarized in Table 1.

### 2.2. Putative TMR–TMR Regulatory Interactions

Under the classical definition, a TMR is a transcription factor that determines cell fate and typically sits at the top of the regulatory hierarchy [9]. At this level, a TMR can control many downstream genes, while relatively few genes regulate its own activity [9,29]. Additionally, TMRs may regulate the expression of other TMRs, creating transcriptional activation cascades that enhance coordinated responses across gene regulatory networks [10].

In this context, our goal was to determine whether, within the inferred regulon network, some TMRs not only regulate their downstream targets but also control other TMRs, positioning them as potential key regulators in the transcriptional circuitry of this neoplasm. After identifying the 50 TMRs that were significant and concordant across the two cohorts (TCGA-THCA/GSE33630), we scanned promoter regions of these 50 TMRs on hg38 using the asymmetric TSS window specified in Methods (−2000/+200 on the positive strand; −200/+2000 on the negative strand). Motifs were available for a subset of TMRs based on public databases, and we used those to search for predicted binding sites among the 50 TMRs, thereby assembling a map of putative TMR–TMR connections (Figure 2). We also examined the differential expression patterns of the 50 TMRs and found directional concordance across datasets in all cases (outer ring, blue/red), reinforcing the biological coherence of their behavior. Finally, to place these predictions in the context of thyroid cancer regulation, motif–promoter hits were filtered to retain only those supported by mutual-information edges from our ARACNe-AP deconvolution, providing an integrated view of how these TMRs may coordinate gene expression programs in PTC.

### 2.3. Functional Relevance of the 50 TMRs

To evaluate the biological importance of the 50 identified TMRs in thyroid cancer, we performed enrichment analysis of the combined regulon network against KEGG pathways and MSigDB Hallmarks (Figure 3). As expected, thyroid hormone synthesis was significantly enriched in the KEGG pathways. We also observed enrichment of valine, leucine, and isoleucine biosynthesis and degradation pathways. Although these amino acids do not directly regulate thyroid function, they are essential for metabolic balance and nutritional health, which can indirectly influence thyroid physiology. Importantly, dysregulation of branched-chain amino acid (BCAA) metabolism has been observed in various cancers [30]. In thyroid cancer, this dysregulation can activate mTORC1 signaling, promoting protein synthesis and cell growth [31]. Metabolomic studies also suggest that BCAAs could be potential biomarkers [32].

Cytoskeletal remodeling and junctional integrity. Actin cytoskeleton remodeling and tight junctions also appeared as enriched categories among genes regulated by the TMR meta-regulon (Figure 3). This supports our previous finding that several TMR families are involved in epithelial–mesenchymal transition (EMT) and metastasis. Both cytoskeletal reorganization and junctional remodeling become dysregulated in cancer and are essential for executing EMT [33], as well as for forming invadopodia that promote invasion and metastatic spread [34].

Estrogen-responsive programs in Hallmarks. Enrichment of MSigDB Hallmarks identified two categories related to estrogen signaling—estrogen response (early) and estrogen response (late) (Figure 3b). It is well established that estrogens act as growth factors in both benign and malignant thyroid cells, activating the MAPK and PI3K pathways, which can coexist with oncogenic mutations such as BRAF mutations or RET/PTC rearrangements [35]. In cell models, estrogen exposure increases VEGF expression, promoting angiogenesis and migration [36], possibly through an ERα/KRT19 axis [37]. Our findings not only support the importance of these pathways but also expand the network of genes and TMRs that may regulate this program in PTC.

### 2.4. Pathway Involvement at the Individual TMR Level

After establishing the global functions of the TMRs, we then evaluated their specific roles within signaling pathways through per-regulon ORA for each of the 50 TMRs. Significant enrichment was found for 18 TMRs in KEGG and for 19 TMRs in the MSigDB Hallmark collection. The results of these individual enrichments are summarized in Figure 4.

#### 2.4.1. The Estrogen Response Is the Most TMR-Rich Program

Among Hallmark pathways, estrogen response showed the highest number of associated TMRs in both early and late phases (Figure 5). For the early estrogen response, we found significant enrichment of nine TMRs: *SMAD9*, *GRHL3*, *MAFB*, *PLAG1*, *ETV5*, *TFCP2L1*, *TEAD4*, *SETBP1*, and *EBF4*. In contrast, during the late estrogen response, the first five regulators—*SMAD9*, *GRHL3*, *MAFB*, *PLAG1*, and *ETV5*—were still present, but *TFCP2L1*, *TEAD4*, *SETBP1*, and *EBF4* were not; instead, *GATAD2A* appeared. This pattern indicates that these five shared TMRs form a core module driving overall estrogen signaling dysregulation, while *TFCP2L1*, *TEAD4*, *SETBP1*, and *EBF4* may mainly influence immediate early responses. Consistent with this, early responses in thyroid models involve inducing *VEGF*, *KRT19*, and activating MAPK/PI3K signaling, while the late phase includes ongoing angiogenesis, survival pathways, and microenvironmental adaptation—features likely regulated by *GATAD2A*.

#### 2.4.2. Four TMRs Dominate Pathway Involvement

*FOXQ1*, *RUNX2*, *SMAD9*, and *GRHL3*, each exhibiting the highest number of enriched categories (four each, Figure 5). In the TMR–TMR promoter analysis, *FOXQ1* is regulated by *ETV5* and forms a reciprocal loop with *GRHL3* (*FOXQ1* ↔︎ *GRHL3*), indicating a feed-forward/feedback module that could position *FOXQ1* above or alongside *GRHL3* in regulating p53 signaling, apical junctions, and early/late estrogen responses. The *RUNX2*-centered regulatory node is overexpressed and, in our analysis, controls KRAS signaling, apical junctions, and coagulation—all categories linked to EMT, which also appears enriched. Notably, in the ARACNe- and motif-supported network, *RUNX2* is regulated by *BHLHE40*, one of the potential central nodes highlighted by our hub analysis. Our analysis also shows that *SMAD9* exhibits significant enrichment for the TGF-β signaling pathway; an intriguing result, given that canonical TGF-β mediators in cancer are SMAD2/3, whereas SMAD1/5/9 usually mediate BMP signaling within the same superfamily [38,39]. This suggests a non-canonical role for *SMAD9* in TGF-β signaling in PTC, thereby expanding the repertoire of transcriptional regulators involved in disease progression. Along these lines, the combined enrichment of estrogen response (early/late) and UV response indicates potential crosstalk among hormonal, genotoxic-stress, and TGF-β superfamily pathways [40,41]. Consistent with its central role, *SMAD9* is regulated by *TEAD4*, another potential central node, in the ARACNe/motif-supported network. Overall, these findings highlight the importance of *SMAD9* beyond its traditional role in BMP signaling, suggesting it may serve as an alternative regulator in thyroid cancer biology.

### 2.5. Top-of-Cascade TMRs

We introduce an objective, quantile-based criterion to prioritize “top-of-cascade” candidates on the ARACNe-supported, motif-backed network: transcription factors with out-degree in the upper quartile (Q3) and in-degree in the lower quartile (Q1). Under this rule, five TFs emerge as robust driver-node candidates—*PBX4* (out/in = 10/3), *GATAD2A* (7/2), *BHLHE40* (7/3), *HEY2* (7/3), and *TEAD4* (7/3). Their excess of outgoing and paucity of incoming connections suggest a high hierarchical position and a potential capacity to initiate or propagate transcriptional programs in PTC.

This is a conservative assessment because the filter requires both sequence-level motif evidence and ARACNe support: (i) not all panel TMRs have available motifs, and (ii) we used a single representative motif per TF, which does not account for all recognition variants of each regulator. As a result, some plausible interactions may be excluded, and TMRs with high global out-degree could fall below the quantile thresholds under these constraints. Below, we analyze each factor by considering network placement, cross-cohort differential expression, regulon-level enrichments, and prior evidence, showing how these nodes could initiate or propagate transcriptional programs in PTC.

#### 2.5.1. PBX Homeobox 4 (PBX4)

*PBX4*, a PBX homeobox transcription factor that acts as an HOX cofactor, has been linked to various tumor types. An extensive pan-cancer study reported widespread dysregulation of PBX4, with overexpression in several cancers but notable underexpression in thyroid carcinoma (THCA), along with associations to stage, methylation, immune infiltration, and pathways connected to MAPK signaling and transcriptional misregulation [42]. In our combined THCA analysis (TCGA-THCA and GSE33630), *PBX4* was also downregulated in tumors compared to normal tissue, consistent with the pan-cancer findings. Although our master-regulator meta-analysis identified other transcription factors as primary network drivers, the consistent repression of *PBX4* in PTC indicates a context-dependent role that should be monitored, including as a potential indicator of pathway rewiring rather than a primary upstream regulator in this disease.

#### 2.5.2. GATA Zinc Finger Domain Containing 2A (GATAD2A)

The network-level signal we observe for *GATAD2A* (p66α) aligns with functional data in thyroid models. In anaplastic thyroid cancer cell lines, shRNA knockdown of *GATAD2A* suppressed proliferation and colony formation, induced G2/M arrest, and markedly increased apoptosis as shown by Annexin V/7-AAD and sub-G1 assays, along with consistent cleavage of caspase-3 and PARP [43]. Moreover, in papillary thyroid cancer (PTC), a circRNA-driven pathway (hsa_circ_0058124/NOTCH3/*GATAD2A*) promotes proliferation, migration/invasion, and in vivo tumor growth; the axis depends on NOTCH3–*GATAD2A* modulation and is proposed as a disease driver in PTC [44]. Mechanistically, *GATAD2A* can also function as a p53 co-activator in breast cancer models by physically interacting with p53, enhancing p53 binding at target promoters (e.g., BAX, NOXA, GADD45A, PAI-1), and suppressing growth and migration. Higher GATAD2A levels are associated with a better prognosis [45]. Although these findings are from breast systems, they support a broader role for *GATAD2A* as a chromatin-linked modulator of tumor suppressive transcriptional programs, consistent with the context-dependent behavior of NuRD subunits.

#### 2.5.3. Basic Helix-Loop-Helix Family Member e40 (BHLHE40)

In our integrated analysis, *BHLHE40*/DEC1 is overexpressed and sits upstream of *RUNX2*, whose regulon converges on KRAS signaling, apical junctions, and coagulation—programs tightly linked to EMT—indicating *BHLHE40* as a coordinator of pro-oncogenic modules in PTC. This placement aligns with thyroid cancer experiments, which show that *BHLHE40* sustains proliferation and invasiveness and functionally cooperates with NOTCH1. Specifically, *BHLHE40* regulates NOTCH1; its overexpression accelerates growth and invasion, while γ-secretase inhibition (DAPT) or NOTCH1 silencing reduces the *BHLHE40*-driven phenotype [46]. Complementing these data, cohort analyses in THCA report *BHLHE40* upregulation associated with immune infiltration (B cells, CD4+ T cells, neutrophils, macrophages, and dendritic cells), and functional evidence indicates that *BHLHE40* knockdown decreases proliferation, migration, invasion, and metastasis traits [47]. Mechanistically, *BHLHE40* can act as either a repressor or an activator, depending on its partners and the chromatin context. In immune and other systems, it modulates cytokine output, cell-cycle control, and metabolism, providing a framework for coupling stress or inflammatory cues to tumor-intrinsic programs in PTC [48].

In anaplastic thyroid carcinoma (ATC), an H19–miR-454-3p–BHLHE40 axis activates PI3K/AKT signaling: H19 sponges miR-454-3p to de-repress *BHLHE40*, and *BHLHE40* overexpression rescues the anti-proliferative and anti-migratory effects of H19 knockdown [49]. Along with the DEC1↔NOTCH1 positive feedback [46], these findings outline two converging routes—NOTCH and H19–miR-454-3p–*BHLHE40* (PI3K/AKT)—through which *BHLHE40* can connect with EMT-related pathways, consistent with our *RUNX2*-focused pathway observations.

#### 2.5.4. Hes-Related Family bHLH Transcription Factor with YRPW Motif 2 (HEY2)

In our integrative analysis of papillary thyroid carcinoma (PTC), *HEY2* was found to be overexpressed, and its regulon was enriched for estrogen-responsive programs, indicating crosstalk between Notch signaling and hormone-responsive circuitry in PTC. Independent functional data in PTC support a pro-tumorigenic role for *HEY2*: the miR-599→*HEY2* axis directly targets the *HEY2* 3′UTR (dual-luciferase), and either overexpressing miR-599 or silencing HEY2 reduces proliferation, migration, invasion, and EMT while promoting apoptosis; these effects are associated with increased E-cadherin/Bax and decreased BCL2, N-cadherin, Vimentin, Snail/Slug, and Notch pathway markers (NOTCH1, DLL1, HES1, NICD, JAG1). Overall, these experiments validate *HEY2* as a Notch-dependent driver and suggest that restoring miR-599 or inhibiting *HEY2* could be a viable therapeutic approach in PTC [50].

Across various other cancers, *HEY2* is often upregulated and linked to aggressive traits, highlighting its pro-oncogenic role. In esophageal squamous cell carcinoma, HEY2 is overexpressed in about 30% of tumors and associates with lymph node metastasis, aligning with Notch pathway activation during disease progression. In non-small-cell lung cancer, a competing endogenous RNA axis involving PRNCR1–miR-448–HEY2 increases HEY2 levels; restoring miR-448 or silencing PRNCR1 reduces proliferation, migration, invasion, and EMT, with partial rescue by *HEY2* re-expression. Mechanistically, HEY proteins (including *HEY1*, *HEY2*, and *HEY*-L) are direct Notch targets that act as bHLH transcriptional repressors through E/N-box binding. This framework explains how Notch→*HEY2* signaling connects developmental or stress signals to EMT-related outcomes—consistent with the estrogen-response enrichment seen in the *HEY2* regulon in PTC [51,52,53,54].

#### 2.5.5. TEA Domain Transcription Factor 4 (TEAD4)

*TEAD4* is a key transcriptional effector of Hippo pathway output, working with YAP/TAZ to regulate programs related to growth, survival, and epithelial plasticity depending on lineage and context. In our integrated PTC analysis, *TEAD4* was under-expressed, indicating a relative decrease in canonical YAP–TEAD signaling. Mechanistically, this matches evidence that the tumor suppressor NF2 (Merlin) directly interacts with *TEAD4* (FERM and C-terminal regions) to inhibit its palmitoylation, decrease nuclear localization, and promote destabilization—thereby limiting *TEAD4*-dependent transcription independently of LATS1/2 and YAP [55]. Additionally, an in vitro thyroid study reports low *TEAD4* expression in tumor tissues and cells, demonstrating that forced *TEAD4* overexpression reduces viability, migration, and invasion, reverses EMT markers (↓N-cadherin, ↓Vimentin; ↑E-cadherin), and influences Wnt signaling (↑WNT3A). Pharmacologic Wnt blockade (IWR-1-endo) negates these effects, supporting a *TEAD4*→Wnt axis that limits pro-metastatic programs in thyroid models [56]. A recent review further places *TEAD4* at the intersection of epithelial plasticity, stress signaling, and therapy response across cancers, while highlighting lineage-specific behaviors [57].

### 2.6. Therapeutic Implications

Based on our meta-analysis, two key, actionable pathways in PTC emerge: (i) a NOTCH3–*GATAD2A* module where *GATAD2A* (p66α) is involved in transcriptional regulation and NuRD-linked chromatin programs, and (ii) a *BHLHE40*–NOTCH dependency that connects stress and inflammatory signals to tumor-inherent growth and invasion. These findings support biomarker-guided strategies aimed at disrupting oncogenic inputs into NOTCH3–*GATAD2A*, restoring *GATAD2A*-driven activation of tumor-suppressing pathways, or targeting vulnerabilities associated with the NuRD complex identified by our TMR subnetwork [44]. Additionally, pharmacologic blockade of NOTCH signaling (such as DAPT) reduces the growth and invasive traits driven by *BHLHE40* overexpression, endorsing therapies that target the *BHLHE40*–NOTCH axis. Since BHLHE40 can act as an activator or repressor by partnering with HDACs, combining modulation of NOTCH signaling with epigenetic regulation presents a promising approach in *BHLHE40*-high PTC cases [46,48].

A third druggable node intersects *TEAD4* biology. TEAD palmitoylation is crucial for TEAD stability and YAP binding, and small-molecule palmitoylation inhibitors or covalent TEAD ligands can reduce YAP/TEAD transcriptional activity [58]. Additionally, the NF2–*TEAD4* interface itself acts as a regulatory chokepoint: *NF2* (Merlin) directly binds to *TEAD4*, inhibiting its palmitoylation, decreasing nuclear residency, and promoting ubiquitin-dependent degradation—thereby limiting *TEAD4* activity independently of LATS1/2 and *YAP* [55]. Given our cohort-specific *TEAD4* downregulation, these mechanisms explain why Hippo pathway constraints may be active in PTC and suggest that restoring *NF2* function or mimicking its inhibitory contact could suppress *TEAD4* where it is oncogenic, while TEAD-axis agents could still target compensatory YAP/TEAD circuitry.

Collectively, these threads support a translational framework that pairs pathway-informed agents with molecular stratification: (a) *NOTCH3*/*GATAD2A*-high or BHLHE40-high tumors prioritized for NOTCH inhibition with or without epigenetic co-targeting; (b) YAP/TEAD-signature–high tumors considered for TEAD-directed agents such as verteporfin (YAP–TEAD disruptor), TED-347 (covalent palmitate-pocket binder), or L06 (auto-palmitoylation inhibitor) [57]. Prospective testing in PTC models should include functional readouts of EMT, migration/invasion, and Wnt/MAPK outputs, and incorporate response biomarkers for the *NOTCH3*–*GATAD2A*, *BHLHE40*–NOTCH/HDAC, and TEAD/NF2 palmitoylation axes to account for context-specific roles and pathway crosstalk intrinsic to thyroid cancer.

Although this study was conducted using transcriptomic data and did not directly integrate variant-level genomic annotations (e.g., OMIM pathogenic variants), several of the transcriptional master regulators identified—such as *FOXE1*, *RUNX2*, *RET*, and *PBX4*—are listed in OMIM as genes harboring monogenic variants associated with congenital thyroid disorders or hereditary cancer susceptibility. However, papillary thyroid carcinoma is generally a multifactorial disease driven by common somatic mutations (e.g., *BRAF*, *RAS*, RET/PTC fusions) and transcriptional reprogramming rather than rare monogenic variants of large effect. Therefore, the master regulators identified in our network are expected to reflect common oncogenic pathways and regulatory perturbations rather than single-gene Mendelian drivers. Future work integrating germline and somatic variant data with regulatory network topology will be essential to further delineate variant-to-regulator causality.

### 2.7. Limitations

This study is primarily computational and theoretical in nature and should be regarded as an integrative systems biology framework rather than definitive experimental validation of transcriptional regulators. As this is a transcriptomic meta-analysis based on only two independent cohorts, traditional statistical heterogeneity tests (e.g., Cochran’s Q, I^2^ metrics) cannot be robustly applied due to insufficient degrees of freedom. To address potential cohort-specific bias, we required strict concordance in effect direction and statistical significance across datasets, and applied Fisher’s meta-analysis with FDR correction to derive consensus transcriptional master regulators. Furthermore, sensitivity analyses were conducted by perturbing network reconstruction parameters, confirming the robustness of key regulators. Still, the limited number of cohorts and lack of functional validation remain important limitations. Therefore, the findings presented here should be interpreted as a regulatory hypothesis-generating framework that lays the foundation for future multi-omics integration and experimental validation in papillary thyroid carcinoma.

While combining two independent cohorts from different platforms (RNA-seq and microarrays) enhances reproducibility, it also imposes conservative filters that may overlook plausible but weaker interactions. The dual-support criterion (motif + ARACNe) further increases specificity at the cost of sensitivity, and motif coverage remains limited by available databases (JASPAR/CIS-BP).

### 2.8. Future Directions

Future studies should test whether silencing pro-oncogenic TMRs such as *GATAD2A*, *BHLHE40*, or *HEY2* reduces proliferation, migration, and EMT in papillary thyroid carcinoma models, while re-expressing downregulated regulators such as *PBX4* or *TEAD4* could restore differentiation and suppress invasion. Perturbation assays in PTC cell lines or organoids, combined with single-cell or spatial transcriptomics, would help clarify causal regulatory relationships and their tissue-specific effects. A summary of expected cellular responses to perturbation of top-of-cascade TMRs is provided in Table 2.

## 3. Materials and Methods

To ensure robustness and avoid single-cohort bias, we analyzed two independent cohorts of papillary thyroid carcinoma, adenocarcinoma, NOS: the TCGA-THCA project and the GEO study GSE33630 [59]. The overall workflow, summarized in Figure 6, combines differential expression analysis, gene set enrichment, regulon inference, and master regulator analysis. The code for this study is available at https://github.com/hachepunto/TiroidesMasterRegulators (accessed on 3 November 2025).

### 3.1. Data Acquisition and Preprocessing

TCGA-THCA. Transcriptomic data were obtained from the Genomic Data Commons (GDC) using the TCGAbiolinks package v. 3.22. We restricted the cohort to samples with a primary diagnosis of Papillary adenocarcinoma, NOS. From raw count files generated by the STAR–Counts workflow, non-protein-coding features were removed, keeping only genes annotated as protein-coding. Duplicate gene symbols were consolidated by applying the statistical mode across expression values. Genes with at least 10 reads in more than 80% of samples were retained, followed by TMM normalization (trimmed mean of M-values) [60] implemented in edgeR [61], and subsequent transformation to CPM (counts per million).

GSE33630 (GEO). Raw “.CEL” files were downloaded from GEO. Affymetrix Human Genome U133 Plus 2.0 arrays were normalized using RMA (Robust Multi-array Average) [62]. Probe annotation was performed with gprofiler2 v. 0.2.3 [63] to map to HGNC symbols, retaining a single identifier per gene by selecting the probe with the highest B-statistic. Genes with low expression (mean log_2_ ≤ 4) and low variability (bottom 25% by variance) were subsequently removed.

### 3.2. Differential Expression Analysis

TCGA-THCA. To identify differentially expressed genes (DEGs) between tumors and normal tissues, we used limma v. 3.66.0 [64] with voom transformation [65] on the CPM-normalized matrix. A linear model was fitted to the tumor versus normal contrast, followed by empirical Bayes variance moderation (eBayes). Genes with adjusted *p* (FDR) < 0.05 and |logFC| ≥ 1 were considered significant.

GSE33630. Differential expression was analyzed with limma, comparing papillary thyroid carcinoma (PTC) directly to normal samples. After Bayesian moderation, genes were retained based on the same criteria (FDR < 0.05 and |logFC| ≥ 1). Results were consolidated into a single identifier per gene, producing a reliable DEG profile for the GEO cohort.

### 3.3. Regulon Network Inference

Regulatory networks were inferred using ARACNe-AP (Algorithm for the Reconstruction of Accurate Cellular Networks, Adaptive Partitioning) [66], based on the curated transcription factor compendium from Lambert et al. (2018) [67].

**Figure 6 ijms-26-11231-f006:**
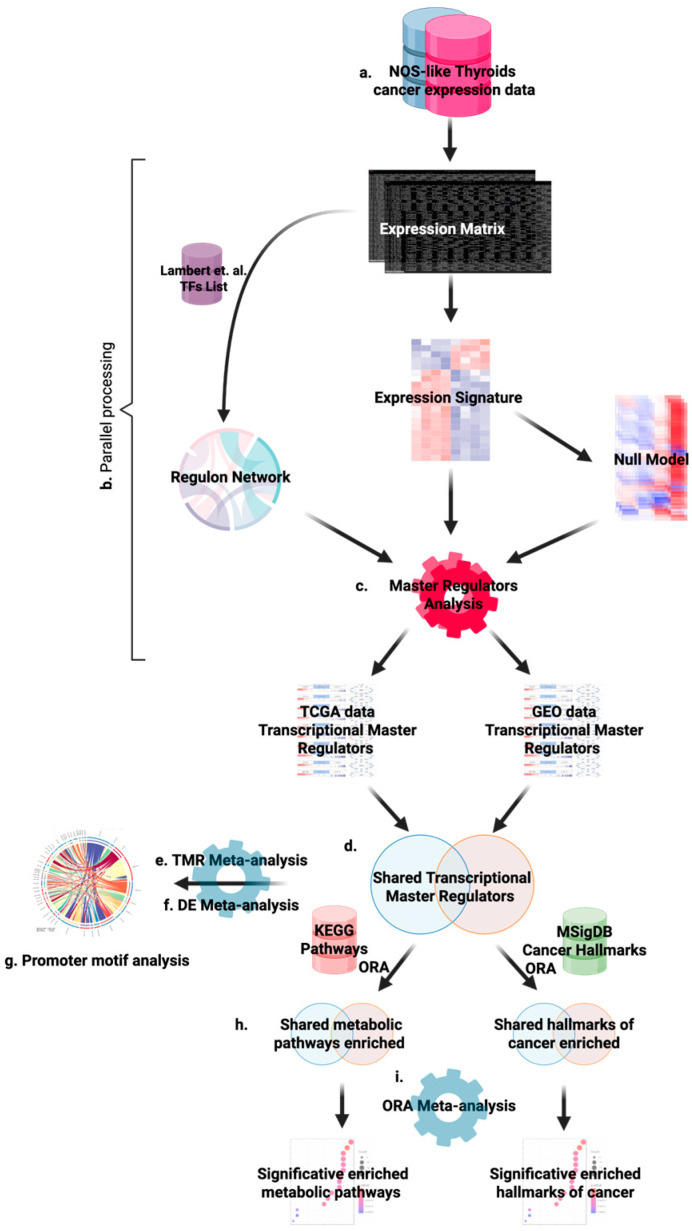
Overview of the analytical pipeline used to identify and functionally characterize transcriptional master regulators (TMRs) in NOS-like papillary thyroid carcinoma. (**a**) Gene expression data from TCGA-THCA (Papillary adenocarcinoma, NOS) and an independent cohort (GSE33630, GEO) are used as inputs. (**b**) ARACNe-AP is applied with the curated human TF catalogue from Lambert et al. (2018) [67] to infer dataset-specific regulon networks. (**c**) Master Regulator Analysis (MRA) is performed with VIPER in each cohort, using tumor-versus-normal signatures and permutation-based null models to estimate TF activity. (**d**) VIPER outputs from both cohorts are integrated via Fisher’s method to define a consensus set of shared TMRs. (**e**,**f**) Meta-analysis results (combined *p*-values and activity metrics) are used to prioritize robust TMRs conserved across datasets. (**g**) Differential expression information for TMRs and their targets is incorporated into a promoter motif–based connectivity analysis: binding motifs (JASPAR2022/TFBSTools) in TMR promoters are intersected with ARACNe edges to build a supported TMR–TMR interaction map. (**h**) In parallel, over-representation analysis (ORA) is performed both on the union of all TMR targets (meta-regulon) and on individual TMR regulons in each cohort using KEGG pathways and MSigDB Hallmarks, with expressed genes as background. (**i**) ORA results are meta-analyzed across cohorts using Fisher’s method and FDR correction to identify biological pathways and cancer hallmarks robustly regulated by the shared TMR architecture.

TCGA-THCA. We used the CPM-normalized expression matrix restricted to protein-coding genes from tumor samples diagnosed as Papillary adenocarcinoma, NOS. The network was generated with a significance threshold of *p* = 1 × 10^−8^, and 100 independent runs with different random seeds were performed, followed by the ARACNe consolidation step to obtain the final network. The resulting network included approximately 234,000 interactions.

GSE33630. Given the smaller sample size (n = 49), we used a more lenient threshold (*p* = 1 × 10^−4^) to account for the reduced statistical power [66,68], employing the same approach of 100 runs and consolidation. The final network contained about 76,000 interactions, consistent with the cohort size.

### 3.4. Master Regulator Analysis (MRA)

Master regulator analysis [13,14] was performed using VIPER (Virtual Inference of Protein-Activity by Enriched Regulon Analysis) v. 1.44.0 implementation [69]. For each dataset, we used the ARACNe-AP-inferred regulon network along with a tumor-versus-normal differential expression signature. VIPER estimates the regulatory activity of each transcription factor by analyzing the enrichment of its targets within the signature and evaluating significance against permutation-based null models. This approach identifies the transcription factors most likely to drive the transcriptional programs (transcriptional master regulators, TMRs) observed in papillary thyroid carcinoma.

### 3.5. Integration of MRA Results by Meta-Analysis (Fisher’s Method)

To identify robust and reproducible master regulators across independent cohorts, we integrated the VIPER MRA *p*-values from TCGA-THCA and GSE33630 using *p*-value meta-analysis. Each dataset contributed one *p*-value per transcription factor, derived from the significance of the VIPER enrichment. We combined *p*-values using Fisher’s method [70], which transforms them into a log-likelihood statistic:(1)$$\chi^{2}=-2\sum_{i=1}^{k}ln(p_{i}),$$
where *p*_*i*_ denotes the *p*-value for the regulator in dataset *i*, and *k* is the number of studies combined (two in our case). Under the null hypothesis of independence, the resulting statistic follows a *χ*^2^ distribution with 2*k* degrees of freedom. We implemented this method using the metap library [71].

This approach enables the detection of transcription factors that are consistent across both studies, even when the signal within an individual dataset is not sufficiently strong. Additionally, we computed the mean NES and recorded sign concordance across cohorts (without using it as an exclusion criterion). The meta-analysis identified 50 TMRs with FDR < 0.05, which constitute the core of this study.

Both datasets were normalized independently to avoid cross-platform artifacts, and replication was evaluated based on the concordance of inferred transcriptional regulators rather than raw expression values, supporting the biological rather than technical origin of the observed consistency.

### 3.6. Motif Analysis of TMRs

Binding motifs were retrieved from JASPAR2022 [72,73,74] as position frequency matrices (PFMs), selecting the latest version for each TF and converting them into position weight matrices (PWMs). We chose JASPAR2022 over newer versions because, at the time of analysis, it was the most recent supported by TFBSTools v. 1.48.0 [75], ensuring reproducibility and compatibility of the workflow. To expand coverage beyond the initial set of JASPAR-only motifs, we added missing TFs with motifs from CIS-BP^33^, standardizing all matrices to PWMs and implementing a source-priority scheme (JASPAR manual > JASPAR automatic > CIS-BP). Metadata for each TF–motif, including source and identifiers, was recorded to ensure reproducibility. For each TMR with an available motif, we scanned the promoters of the 50 TMRs on hg38 using an asymmetric TSS window (−2000/+200 on the positive strand; −200/+2000 on the negative strand) with min.score = 85%. To minimize false positives, we retained only motif–target pairs supported by the ARACNe-AP network in at least one dataset, combining sequence evidence with coexpression/MI support.

### 3.7. Differential Expression of TMRs via Meta-Analysis (Fisher + IVW Fixed Effect)

For the 50 validated TMRs, we evaluated whether they show consistent changes in expression between tumors and controls by integrating evidence from both cohorts.

Combining significance levels. For each TMR, differential-expression *p*-values from each dataset were combined using Fisher’s method.(2)$$\chi^{2}=-2\sum_{i=1}^{k}ln(p_{i}),\;\;\chi^{2}∼\chi_{2k}^{2},$$
(*k* = 2), followed by multiple-testing correction (FDR).

Combining effect sizes (logFC). Complementarily, we combined log fold-changes using a common-effect (fixed-effect) inverse-variance–weighted (IVW) meta-analysis [76,77]:(3)$${\hat{\beta }}_{meta}=\frac{\Sigma_{i}w_{i}\hat{\beta_{i}}}{\Sigma_{i}w_{i}},\;w_{i}=\frac{1}{{SE}_{i}^{2}},\;\;Z=\frac{{\hat{\beta }}_{meta}}{\sqrt{\frac{1}{\Sigma_{i}w_{i}}}},$$
yielding a two-sided *p*-value for ${\hat{\beta }}_{meta}$ that was subsequently FDR-adjusted. Standard errors for each cohort were estimated from limma outputs (*t*-statistic and logFC).

Finally, we verified directional concordance of the effect (sign of logFC) across datasets. This Fisher + IVW scheme jointly assesses the significance and magnitude of change, improving robustness to cohort-specific variability.

### 3.8. Functional Enrichment of the Meta-Regulon

To characterize the biological processes collectively controlled by the identified TMRs, we created a meta-regulon defined as the union of targets regulated by the 50 validated TMRs, combining regulons from both datasets. Over-representation analyses (ORA) were performed on this set for MSigDB Hallmarks [78], KEGG, and GO, for each dataset, using the corresponding universe of expressed genes as background. Term-level *p*-values were then combined across cohorts using Fisher’s method and adjusted for FDR. This layer summarizes the biological programs shared by the combined network of TMR targets.

### 3.9. Sensitivity and Robustness Analyses

To evaluate the robustness of the identified transcriptional master regulators (TMRs) and minimize the likelihood of false-positive results, we performed a multi-level sensitivity analysis integrating network- and regulator-level evaluations (Appendix A).

#### 3.9.1. Network-Level Robustness

ARACNe-AP networks were reconstructed using three mutual information thresholds (*p* = 1 × 10^2^, 1 × 10^4^, and 1 × 10^8^) and three bootstrap depths (50, 100, and 200). As shown in Appendix A, TMR sets were highly consistent across all reconstructions. Within each dataset, the overlap among bootstrap-derived networks exceeded 90% (Jaccard = 0.75–0.91; Spearman’s = 0.97–0.99), while across p-cutoffs the overlap reached 95–100% (Jaccard = 0.67–0.89). These results indicate strong structural invariance of the inferred regulons.

#### 3.9.2. Regulator-Level Stability

We next assessed the persistence of individual TMRs across all 18 analyses (Appendix A). Core regulators such as *SMAD9*, *PRDM16*, *EBF4*, *RUNX2*, and *BHLHE40* were consistently significant (*p* < 0.05 in 100% of runs) and ranked among the top 5–50 transcription factors by absolute NES (median percentile rank ≥ 94%). Additional regulators including *TFCP2L1*, *TEAD4*, *FOXQ1*, and *CREB5* remained significant in ≥83% of runs, reflecting robust though slightly parameter-dependent effects.

#### 3.9.3. Cross-Cohort Reproducibility

Effect-size meta-analysis confirmed consistent directionality and statistical significance across TCGA-THCA and GSE33630 datasets (empirical *p* < 110), supporting cross-cohort reproducibility of the regulatory signals.

Collectively, these results demonstrate that the TMRs identified in this study are highly reproducible across multiple parameter spaces and sampling perturbations, and are therefore unlikely to represent random or dataset-specific false positives.

## 4. Conclusions

Functionally, estrogen response became a key program, with strong Hallmark enrichment observed across cohorts. The early and late components were supported by nine TMRs—*SMAD9*, *GRHL3*, *MAFB*, *PLAG1*, *ETV5*, *TFCP2L1*, *TEAD4*, *SETBP1*, and *EBF4*—highlighting a hormonal axis likely important to PTC initiation and progression.

Regulon-level analyses further implicated *SMAD9* in TGF-β signaling, indicating non-canonical activity (usually associated with BMP) and possible crosstalk with estrogen and genotoxic stress pathways. Along with the edges *TEAD4* → *SMAD9* and *BHLHE40* → *RUNX2*, these findings highlight *SMAD9*—and the quantile-prioritized TMR set—as key regulators with potential functional and therapeutic significance in PTC.

## Figures and Tables

**Figure 1 ijms-26-11231-f001:**
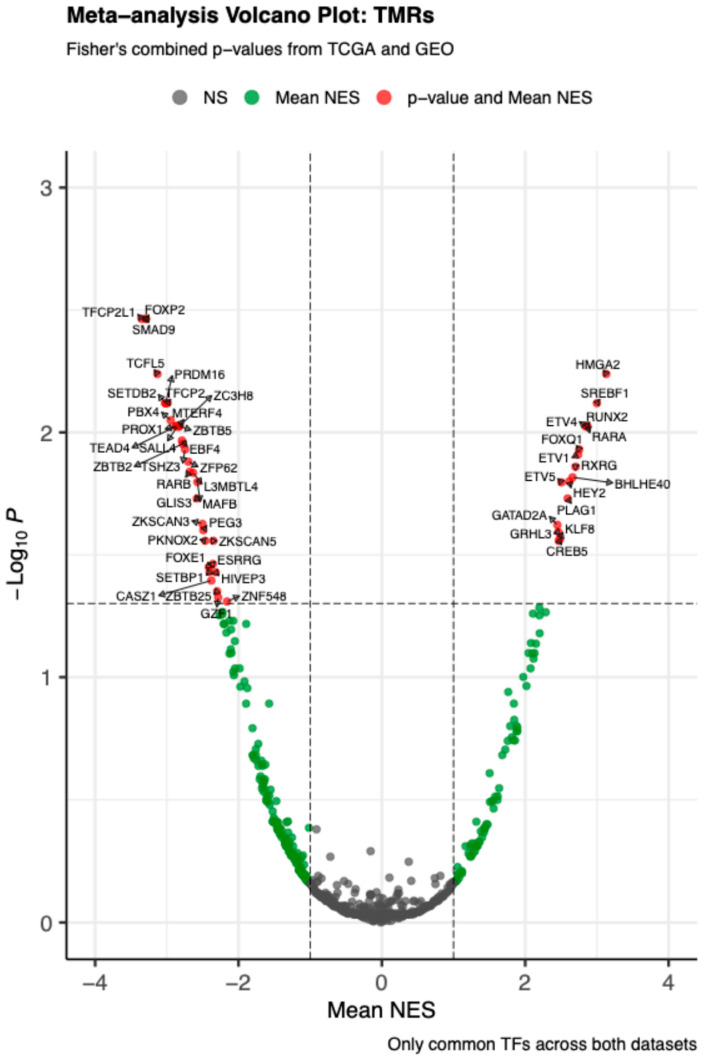
Meta–analysis volcano plot of transcriptional master regulators (TMRs). The volcano plot shows the mean normalized enrichment score (Mean NES, x–axis) versus the –log10 transformed Fisher’s combined *p*-value (y–axis) across TCGA and GEO datasets. Each point represents a transcription factor (TF). Vertical dashed lines indicate the cutoff for |Mean NES| > 1, and the horizontal dashed line marks a meta-adjusted *p*-value of 0.05. Gray points denote non-significant TFs, green points indicate those significant for mean NES only, and red points represent TFs significant for both mean NES and meta-adjusted *p*-value. Labels highlight selected significant TFs.

**Figure 2 ijms-26-11231-f002:**
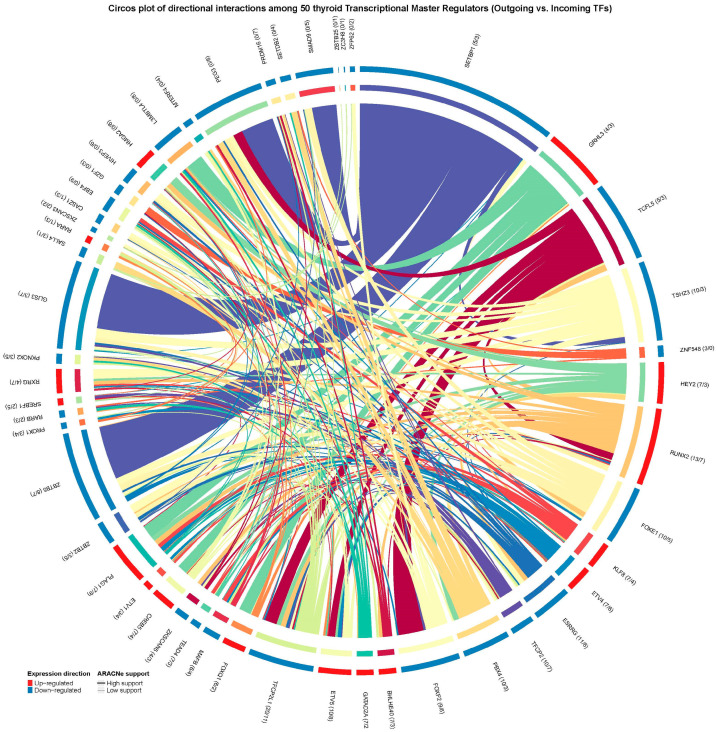
Circos plot showing directional interactions among the 50 transcriptional master regulators identified in NOS-like thyroid cancer. Each arc represents a TMR, uniquely colored for easy visual identification. Outgoing links indicate predicted regulatory targets (arrows), while incoming links show regulators acting on that TMR. Arc halos demonstrate the consensus direction of differential expression across TCGA and GEO datasets (red = up-regulated, blue = down-regulated). Links are colored according to their originating TMR and weighted by motif support, with additional annotation indicating ARACNe mutual information support (thick, saturated = high support; thin, lighter = low support). Numbers in parentheses beside each TMR specify the number of outgoing versus incoming supported interactions (targets/regulators). This integrated visualization highlights candidate regulatory hubs and their coordinated activity in thyroid tumorigenesis.

**Figure 3 ijms-26-11231-f003:**
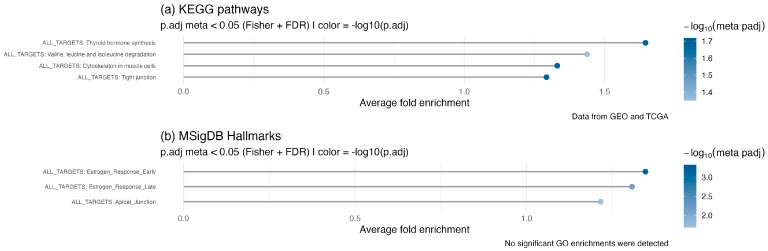
Meta-analysis of functional enrichment across the combined targets of the 50 TMRs. Over-representation analysis (ORA) results for (**a**) KEGG pathways and (**b**) MSigDB Hallmarks. Only pathways significantly enriched in both datasets (GEO and TCGA) were retained, and *p*-values were combined using Fisher’s method followed by false discovery rate (FDR) correction. The x-axis represents the average fold enrichment across datasets, while color intensity indicates the −log_10_ of the FDR-adjusted meta *p*-value. No significant enrichments were detected in Gene Ontology (GO) terms.

**Figure 4 ijms-26-11231-f004:**
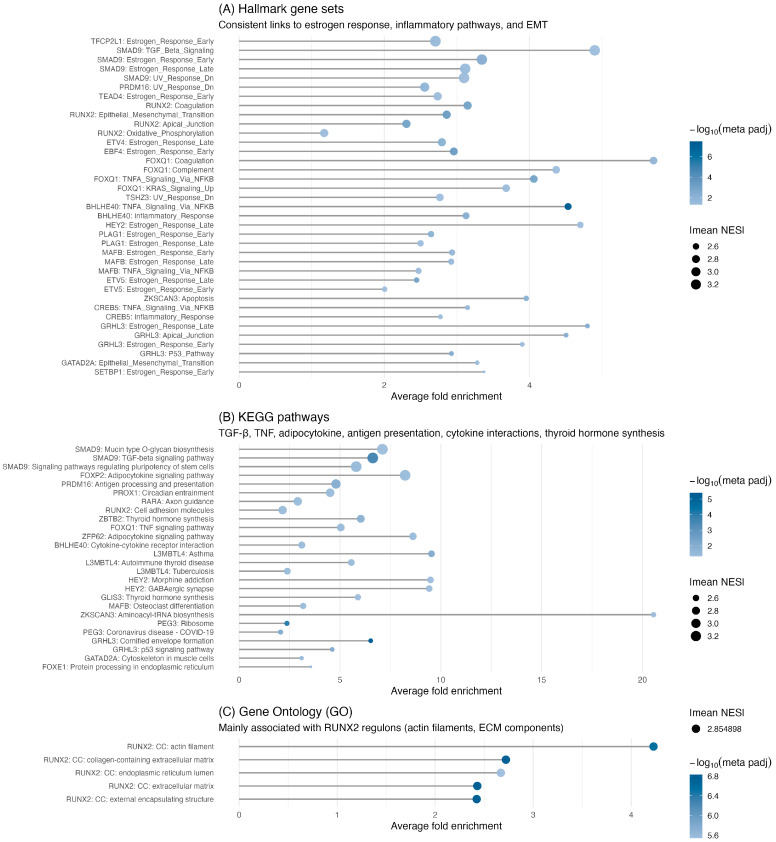
Meta-analysis of functional enrichment across TMR regulons. Panels show ORA results for regulons identified in both TCGA and GEO datasets. (**A**) Hallmark gene sets reveal consistent enrichment in estrogen response, inflammatory, and epithelial–mesenchymal transition pathways. (**B**) KEGG pathways highlight signaling cascades such as TGF-β, TNF, and adipocytokine signaling, as well as immune processes including antigen presentation and cytokine interactions, and thyroid hormone synthesis. (**C**) Gene Ontology (GO) categories are primarily associated with *RUNX2* regulons, including actin filament and extracellular matrix organization. The x-axis displays the average fold enrichment across datasets, point size represents |mean NES|, and color intensity reflects the −log_10_ of the FDR-adjusted meta *p*-value (Fisher’s method, FDR < 0.05).

**Figure 5 ijms-26-11231-f005:**
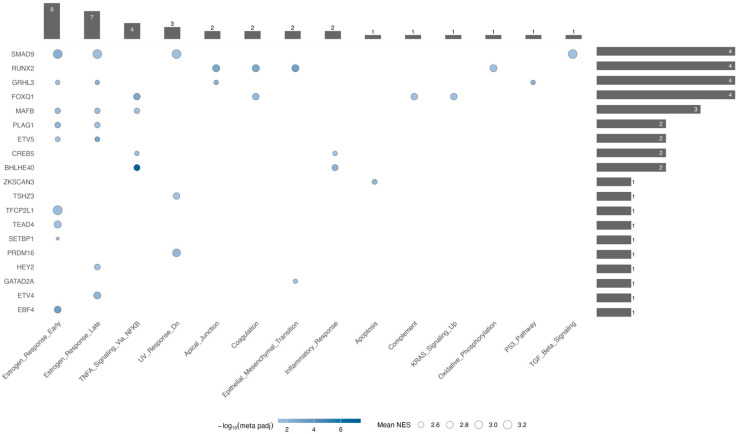
Hallmark pathway enrichment across TMR regulons. Bubble-style upset-like heatmap showing the overlap between transcriptional master regulators (TMRs, y-axis) and enriched hallmark pathways (x-axis) identified by meta-analysis of ORA across GEO and TCGA datasets. Bubble color represents the −log_10_ of the meta-adjusted *p*-value, and bubble size indicates the mean normalized enrichment score of each TMR. Vertical bars indicate the number of hallmarks enriched per TMR, while horizontal bars represent the number of TMRs associated with each hallmark. Notably, the top four TMRs (*SMAD9*, *RUNX2*, *GRHL3*, and *FOXQ1*) collectively account for 11 out of 13 enriched hallmarks (~85%), suggesting that a limited subset of regulators drives most hallmark-associated transcriptional programs in thyroid cancer.

**Table 1 ijms-26-11231-t001:** Transcription factor families of the 50 transcriptional master regulators (TMRs) identified in papillary thyroid carcinoma (PTC).

Transcription Factor Family	TMRs
Zinc Finger	*ZBTB2*, *ZBTB25*, *ZBTB5*, *ZKSCAN3*, *ZKSCAN5*, *ZFP62*, *ZNF548*, *ZC3H8*
Forkhead (FOX)	*FOXE1*, *FOXP2*, *FOXQ1*
Basic Helix-Loop-Helix (bHLH)	*BHLHE40*, *HEY2*
ETS	*ETV1*, *ETV4*, *ETV5*
Nuclear Receptors (NR)	*ESRRG*, *RARA*, *RARB*, *RXRG*
Kruppel-like/C2H2 zinc finger-related	*KLF8*
Sall	*SALL4*
PRDM	*PRDM16*
SMAD	*SMAD9*
TEAD	*TEAD4*
TFCP	*TFCP2*, *TFCP2L1*
PLAG	*PLAG1*
PROX	*PROX1*
RUNX	*RUNX2*
SREBF	*SREBF1*
PBX	*PBX4*
MAF	*MAFB*
GRHL	*GRHL3*
GLIS	*GLIS3*
Others/Unclassified	*CASZ1*, *CREB5*, *GATAD2A*, *GZF1*, *HMGA2*, *L3MBTL4*, *MTERF4*, *PEG3*, *PKNOX2*, *TCF15*, *TSHZ3*

**Table 2 ijms-26-11231-t002:** Proposed functional hypotheses for top-of-cascade TMRs identified in PTC.

TMR	Knockdown—Expected Response	Overexpression—Expected Response
*PBX4*	Enhanced proliferation and dedifferentiation; possible activation of MAPK and cell-cycle genes.	Restoration of differentiated phenotype; reduced proliferation and invasiveness.
*GATAD2A*	Reduced proliferation and migration; increased apoptosis through p53-related pathways.	Increased proliferation, migration, and resistance to apoptosis.
*BHLHE40*	Decreased EMT and invasiveness; reduced expression of NOTCH and PI3K/AKT targets.	Promotion of EMT and inflammatory response; increased invasiveness.
*HEY2*	Reversal of EMT (↑ E-cadherin, ↓ Vimentin/Snail); decreased proliferation and migration.	Induction of EMT and survival signaling via the NOTCH pathway.
*TEAD4*	Further loss of epithelial traits and enhanced migration (context-dependent).	Restoration of adhesion and epithelial markers; reduced EMT and invasiveness.

## Data Availability

The data used in this study were derived from public domain resources, including the Gene Expression Omnibus (GSE33630, https://www.ncbi.nlm.nih.gov/geo/query/acc.cgi?acc=GSE33630 accessed on 3 November 2025) and The Cancer Genome Atlas Thyroid Cancer dataset (TCGA-THCA, https://doi.org/10.7937/K9/TCIA.2016.9ZFRVF1B accessed on 3 November 2025). Analysis scripts and reproducible code are available at https://github.com/hachepunto/TiroidesMasterRegulators accessed on 3 November 2025.

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
