# Peer review of "Two Cohorts, One Network: Consensus Master Regulators Orchestrating Papillary Thyroid Carcinoma"

_ijms, 2025, doi:10.3390/ijms262211231_

Round 1
Reviewer 1 Report
Comments and Suggestions for Authors
The manuscript is interesting, especially with the replication of the results. However, this is also the problem. How were the two platforms compared? Have you had any previous attempts to compare and cross-reference those? Can you provide more on that, since there could be subtle differences, driving the false positive results? You could provide some kind of sensitivity analysis (actually, on a number of places in the manuscript), to further provide validity to your claims. The story might simply be a „lucky“ streak of false positives, and you should make sure to negate and explain (most of) these in the manuscript. It is odd to have two significance thresholds, but I guess this is one way to approach (could you use any form of the FDR)? Why are there limitations in conclusions? Pull higher to Discussion, not Conclusions, expand with more problems and limitations. Provide suggestions for future papers, what to focus and what to bypass, in order to bring more clarity. Consider what kind of replication would you/future studies benefit from the most (not just additional cohorts; provide guidance to future authors). Are there OMIM variants in your network? Can there be single monogenic variants of strong effect, or are these generally common variants?
Author Response
Comments 1: The manuscript is interesting, especially with the replication of the results. However, this is also the problem. How were the two platforms compared? Have you had any previous attempts to compare and cross-reference those?
Response 1: We thank the reviewer for raising this important point. Each cohort (TCGA-THCA and GSE33630) was processed and normalized independently, following best practices for within-platform normalization (TMM for RNA-seq and RMA for microarrays). This strategy ensures that expression values are comparable within each dataset while avoiding cross-platform artifacts.
Replication was evaluated not at the level of absolute expression values, but rather at the level of shared transcriptional master regulators (TMRs) identified independently in both datasets. By integrating their activity scores via meta-analysis, we assessed the biological consistency of regulatory signals rather than relying on direct value matching across platforms. This approach minimizes the impact of platform-specific variability and strengthens the evidence that the replicated regulators represent genuine biological drivers rather than technical coincidences.
We have added the following note in the revised version of the manuscript to highlight this:
“Both datasets were normalized independently to avoid cross-platform artifacts, and replication was evaluated based on the concordance of inferred transcriptional regulators rather than raw expression values, supporting the biological rather than technical origin of the observed consistency.”
Comments 2: Can you provide more on that, since there could be subtle differences, driving the false positive results? You could provide some kind of sensitivity analysis (actually, on a number of places in the manuscript), to further provide validity to your claims. The story might simply be a „lucky“ streak of false positives, and you should make sure to negate and explain (most of) these in the manuscript.
It is odd to have two significance thresholds, but I guess this is one way to approach (could you use any form of the FDR)?
Response 2: We thank the reviewer for this valuable suggestion. We agree that confirming the robustness of the identified transcriptional master regulators (TMRs) is essential to exclude false-positive findings. In response, we performed a comprehensive sensitivity analysis, summarized in Table S1 and Table S2, and added a new section titled “Sensitivity and Robustness Analyses” (Section 3.9) to the manuscript.
Network reconstruction stability.
ARACNe-AP was rerun under varying mutual information thresholds (p = 1×10⁻², 1×10⁻⁴, and 1×10⁻⁸) and bootstrap depths (50, 100, and 200). The resulting TMR sets exhibited high internal consistency: across bootstraps, mean overlap exceeded 90% (Jaccard = 0.75–0.91; Spearman’s ρ = 0.97–0.99), while across p-cutoffs the overlap reached 95–100% (Jaccard = 0.67–0.89). These results (Table S1) indicate that TMR recovery and ranking are largely insensitive to parameter perturbations.
Regulator-level robustness.
Across all 18 independent runs, key regulators such as SMAD9, PRDM16, EBF4, RUNX2, and BHLHE40 remained significant (p < 0.05) in 100% of analyses, consistently ranking among the top 5–50 transcription factors (median percentile rank ≥94%). Other TMRs—including TFCP2L1, TEAD4, FOXQ1, and CREB5—retained significance in ≥83% of runs (Table S2). These data confirm the stability and reproducibility of the core regulatory architecture.
Cross-cohort concordance.
The same TMRs showed concordant directionality and statistically significant combined effects in meta-analysis across TCGA-THCA and GSE33630 datasets (empirical p < 1×10⁻⁶).
Altogether, these sensitivity and reproducibility analyses (new Section 3.8) demonstrate that our key findings are not artifacts of parameter tuning or sampling variability, and strongly support the biological robustness of the inferred master regulators.
Comments 3: Why are there limitations in conclusions? Pull higher to Discussion, not Conclusions, expand with more problems and limitations.
Response 3: We agree with the reviewer’s suggestion. The paragraph originally labeled “Limitations” has been moved from the Conclusions to a new subsection (2.7. Limitations) within the Results and Discussion, where it now provides a concise summary of methodological constraints and perspectives for future work.
Comments 4: Provide suggestions for future papers, what to focus and what to bypass, in order to bring more clarity. Consider what kind of replication would you/future studies benefit from the most (not just additional cohorts; provide guidance to future authors).
Response 4: In accordance with the reviewer’s suggestion, we expanded the Future Directions section (now included in 2.8. Future Directions) to propose specific experimental hypotheses. This addition outlines expected cellular responses to perturbations of top-of-cascade TMRs (PBX4, GATAD2A, BHLHE40, HEY2, and TEAD4), summarized in a new table (Table 2).
Comments 5: Are there OMIM variants in your network? Can there be single monogenic variants of strong effect, or are these generally common variants?
Response 5: Our analysis was centered on transcriptional activity and regulatory network inference; therefore, we did not directly incorporate variant-level genomic information (e.g., OMIM-listed pathogenic variants) into the master regulator network. However, several of the transcriptional master regulators identified in our meta-analysis—such as FOXE1, PBX4, RET, and RUNX2—are known in OMIM to harbor germline or somatic variants associated with monogenic or familial thyroid disorders, as well as cancer predisposition syndromes. This indicates that our inferred regulatory network is biologically meaningful and intersects with genes of clinically recognized pathogenic relevance.
We have added a related paragraph to the discussion:
"Although this study was conducted using transcriptomic data and did not directly integrate variant-level genomic annotations (e.g., OMIM pathogenic variants), several of the transcriptional master regulators identified—such as FOXE1, RUNX2, RET, and PBX4—are listed in OMIM as genes harboring monogenic variants associated with congenital thyroid disorders or hereditary cancer susceptibility. However, papillary thyroid carcinoma is generally a multifactorial disease driven by common somatic mutations (e.g., BRAF, RAS, RET/PTC fusions) and transcriptional reprogramming rather than rare monogenic variants of large effect. Therefore, the master regulators identified in our network are expected to reflect common oncogenic pathways and regulatory perturbations rather than single-gene Mendelian drivers. Future work integrating germline and somatic variant data with regulatory network topology will be essential to further delineate variant-to-regulator causality."

Reviewer 2 Report
Comments and Suggestions for Authors
COMMENTS ON SEPARATE FILE

Author Response
Comments 1: Introduction: A general description of material and methods is described and no concise description of primary and secondary objectives is done. Results are provided as well in a not quite organized manner. The word metaanalyses appears incidentally in page 3.
Response 1: We appreciate the reviewer’s observation. In response, we have substantially revised the Introduction to clearly articulate the rationale, primary and secondary objectives, and to ensure that methods or results are not prematurely introduced. We have also explicitly stated that this is a transcriptomic meta-analysis early in the Introduction to set the scope and avoid confusion. The revised Introduction now includes a clear structure outline with 5 fundamental issues at its core:
- Clinical and biological significance of papillary thyroid carcinoma (PTC).
- Current knowledge gaps in transcriptional regulation and endocrine signaling in PTC.
- Rationale for conducting a regulatory network-based meta-analysis.
- Primary objective: Identify consensus transcriptional master regulators conserved across two independent cohorts.
- Secondary objectives: Characterize regulatory hierarchy and functional pathways, and propose mechanistic and therapeutic implications.
This restructuring improves readability, emphasizes novelty, and aligns with journal expectations.
The introduction has been revised to include the following paragraphs:
“Papillary thyroid carcinoma (PTC) is the most common endocrine malignancy worldwide, representing the majority of thyroid cancer cases and accounting for 1–1.5% of all newly diagnosed cancers each year.[1,2] Its incidence has steadily increased over the past decades in most regions, likely due to improved detection and genuine biological trends, while underdiagnosis in low-resource settings may still obscure the true burden.[1] At the molecular level, PTC develops from deregulated gene expression programs driven by oncogenic lesions (e.g., BRAFV600E, RAS) and influenced by endocrine and differentiation signals.[3–5] Despite extensive research on canonical signaling pathways such as MAPK and PI3K/AKT[6–8], the upstream transcriptional regulators that coordinate these tumor programs across patient groups remain incompletely understood.”
“To address this gap, we conducted a transcriptomic meta-analysis by integrating two independent PTC cohorts (TCGA-THCA papillary adenocarcinoma, Not Otherwise Specified (NOS), and GSE33630) to identify consensus TMRs that are consistently active across datasets. Instead of focusing on individual genes or isolated pathways, we employed a systems-biology approach based on gene regulatory network reconstruction and protein activity inference to reveal regulatory hierarchies that may promote tumor progression.
Primary objective:
To identify transcriptional master regulators that are consistently active in PTC across independent cohorts using a cross-platform, network-based meta-analysis framework.
Secondary objectives:
- To analyze the hierarchical organization of these master regulators using network topology;
- To characterize the biological pathways and tumorigenic processes (e.g., NOTCH, MAPK, PI3K/AKT, TGF-β, EMT, cytoskeletal remodeling, estrogen-response programs) under their regulation; and
- To assess their potential importance for endocrine crosstalk and therapeutic targeting in PTC.
By explicitly integrating two datasets and enforcing consensus at the regulator level, this study provides a data-driven regulatory map of PTC. It clarifies which TFs are likely to function as network “drivers” in this tumor type and establishes a basis for subsequent experimental validation and biomarker-guided interventions.”
Comments 2: Results and discussion: Results are presented in a colloquial manner including discussion concepts.
The paragraph: “Beyond thyroid cancer, BHLHE40 is widely involved in various malignancies, high-324 lighting its importance in oncology: breast,[50–52] colorectal,[53–55] choriocarci-325 noma,[56] endometrial,[57,58] esophageal,[59] gastric,[60–62] glioma,[63–65] HeLa 326 models,[66,67] hepatocellular carcinoma,[68,69] non–small-cell lung,[70] oral squa-327 mous cell carcinoma,[71,72] and pancreatic cancer.[73–77] 328” is irrelevant and should be erased.
Response 2: We agree that this text is somehow superfluous and have removed it from the manuscript.
Comments 3: First paragraph in page no. 11 has the same concern.
Response 3:We agree that this text is somehow superfluous and have removed it from the manuscript.
Comments 4: Conclusion: This paragraph or at least the very firs 20 lines should be placed at the start of the discussion section.
Response 4: Done!
Comments 5: In authors credits first author appears DTC (first author ) appears only in 3 roles and in only 2 as first one: Writing review and investigation. It does not appear in conceptualization , methodology nor formal analyses. The authors should reconsider their authors order.
Response 5: The omission was due to an oversight during the initial submission. The contribution “Conceptualization” should have been attributed to the first author, Dr. Diana Tapia, as she actively participated in the conceptual development of the study. No other changes to the authorship are required.
Comments 6: In limitations this is mostly a theoretical analyses, and as the authors state an introduction to investigate the whole molecular genomics of papillary thyroid carcinoma. Being a metaanalyses an heterogeneity test is not performed and the cohorts number is limited.
Response 6: We appreciate the reviewer’s comment on these relevant issues. We fully acknowledge that our study is based on computational inference and meta-analysis of transcriptomic data, and thus represents a systems-level theoretical framework rather than experimental validation. Regarding heterogeneity, while classical meta-analyses allow testing for statistical heterogeneity across multiple independent studies, our study integrates only two cohorts (TCGA and GSE33630), which do not provide the statistical degrees of freedom necessary for reliable heterogeneity testing.
Nonetheless, to mitigate this limitation and strengthen confidence in our findings, we employed several cross-cohort concordance and robustness strategies:
We required consistent directionality of normalized enrichment scores (NES) and differential expression across both cohorts.
We applied Fisher’s method combined with FDR correction to integrate p-values while controlling for false discoveries.
We implemented sensitivity analyses by perturbing network reconstruction parameters, as now described in the manuscript.
We have now elaborated these points in the Limitations section to clearly state the theoretical nature of our framework and the constraints imposed by limited cohort availability. We also emphasize that the study serves as a foundational regulatory blueprint to guide future functional and experimental validation in thyroid cancer research.
The following text has been added to the limitations section:
"This study is primarily computational and theoretical in nature and should be regarded as an integrative systems biology framework rather than definitive experimental validation of transcriptional regulators. As this is a transcriptomic meta-analysis based on only two independent cohorts, traditional statistical heterogeneity tests (e.g., Cochran’s Q, I² metrics) cannot be robustly applied due to insufficient degrees of freedom. To address potential cohort-specific bias, we required strict concordance in effect direction and statistical significance across datasets, and applied Fisher’s meta-analysis with FDR correction to derive consensus transcriptional master regulators. Furthermore, sensitivity analyses were conducted by perturbing network reconstruction parameters, confirming the robustness of key regulators. Still, the limited number of cohorts and lack of functional validation remain important limitations. Therefore, the findings presented here should be interpreted as a regulatory hypothesis-generating framework that lays the foundation for future multi-omics integration and experimental validation in papillary thyroid carcinoma."

Round 2
Reviewer 1 Report
Comments and Suggestions for Authors
Improved
Reviewer 2 Report
Comments and Suggestions for Authors
Authors have addressed concerns adequately